# Epigenomic and Transcriptomic Prioritization of Candidate Obesity-Risk Regulatory GWAS SNPs

**DOI:** 10.3390/ijms23031271

**Published:** 2022-01-23

**Authors:** Xiao Zhang, Tian-Ying Li, Hong-Mei Xiao, Kenneth C. Ehrlich, Hui Shen, Hong-Wen Deng, Melanie Ehrlich

**Affiliations:** 1Tulane Center for Biomedical Informatics and Genomics, Division of Biomedical Informatics and Genomics, Deming Department of Medicine, School of Medicine, Tulane University, New Orleans, LA 70112, USA; xzhang30@tulane.edu (X.Z.); kehrlich@tulane.edu (K.C.E.); hshen3@tulane.edu (H.S.); 2Center for System Biology, Data Sciences, and Reproductive Health, School of Basic Medical Science, Central South University, Changsha 410013, China; ltynothing@163.com (T.-Y.L.); hmxiao@csu.edu.cn (H.-M.X.); 3Tulane Cancer Center and Hayward Genetics Center, Tulane University, New Orleans, LA 70112, USA

**Keywords:** GWAS, epigenetics, obesity, SNPs, lncRNA, RNA isoforms, enhancers, promoters, transcriptomics, subcutaneous adipose tissue

## Abstract

Concern about rising rates of obesity has prompted searches for obesity-related single nucleotide polymorphisms (SNPs) in genome-wide association studies (GWAS). Identifying plausible regulatory SNPs is very difficult partially because of linkage disequilibrium. We used an unusual epigenomic and transcriptomic analysis of obesity GWAS-derived SNPs in adipose versus heterologous tissues. From 50 GWAS and 121,064 expanded SNPs, we prioritized 47 potential causal regulatory SNPs (Tier-1 SNPs) for 14 gene loci. A detailed examination of seven loci revealed that four (*CABLES1*, *PC*, *PEMT*, and *FAM13A*) had Tier-1 SNPs positioned so that they could regulate use of alternative transcription start sites, resulting in different polypeptides being generated or different amounts of an intronic microRNA gene being expressed. *HOXA11* and long noncoding RNA gene *RP11-392O17.1* had Tier-1 SNPs in their 3′ or promoter region, respectively, and strong preferences for expression in subcutaneous versus visceral adipose tissue. *ZBED3-AS1* had two intragenic Tier-1 SNPs, each of which could contribute to mediating obesity risk through modulating long-distance chromatin interactions. Our approach not only revealed especially credible novel regulatory SNPs, but also helped evaluate previously highlighted obesity GWAS SNPs that were candidates for transcription regulation.

## 1. Introduction

The rapidly rising rates of obesity are a major concern for public health and increased risk of disease, including type 2 diabetes (T2D), cardiovascular disease, osteoarthritis, certain kinds of cancer, and respiratory problems [1,2]. Adipose tissue expansion involves an increase in the number of adipocytes and/or in the size of adipocytes. However, not only overall obesity, but also the location of the fat depot with increased adipose tissue is of importance to health risk [2]. The major types of fat depots are visceral adipose tissue (VAT), which is deposited around internal organs in the abdominal cavity, and subcutaneous adipose tissue (SAT), which is beneath the skin. High levels of VAT are associated with metabolic abnormalities while high levels of SAT may be protective rather than harmful [3]. Similarly, adipose deposition around the hips, which mostly involves SAT, is less likely to be a risk factor for various chronic diseases than are high levels of adipose at the waist, which have a large VAT component. Pre-menopausal women with their tendency for high SAT/VAT ratios are at lower risk of adipose-related disease [4]. Nonetheless, expansion of SAT depots may also contribute to health either in its protective role or through its interactions with VAT depots [3,5].

The clinical relevance of the location of fat depots in obesity led to more detailed measurements of obesity than just body mass index (BMI), namely, the waist-to-hip ratio (WHR) adjusted for BMI (WHR_adjBMI_), waist circumference adjusted for BMI (WC_adjBMI_), and hip circumference adjusted for BMI (HC_adjBMI_). These anthropomorphic measures are the most frequent ones used in genome-wide association studies of the genetic risk of obesity (obesity GWAS) because of their ease of measurement and, for the last three measures, their health-related emphasis on the distribution of fat depots [3]. Genetic risk clearly contributes to obesity, and genetic heritability for BMI has been estimated to be 30–40% [2], but this estimate depends on the study design [6].

Especially challenging in the analysis of most GWAS-derived genetic variants is assigning a biochemical function to them when they are not both exonic and predicted to change the encoded protein’s activity or stability. For example, less than 10% of single nucleotide polymorphisms (SNPs, the most frequent genetic variants) significantly associated with obesity in GWAS are in coding regions [7], and only a fraction of these SNPs is in non-synonymous codons predicted to affect polypeptide structure. Most of the GWAS-derived SNPs that actually mitigate genetic risk for a given trait are probably in enhancers or, less frequently, promoters [8]. The detection of regulatory SNPs is complicated by the variety of sizes and genomic locations for cell-type specific enhancers. Especially difficult is the finding that there are often tens to hundreds of SNPs tightly linked to an individual GWAS-reported SNP, a problem that also is characteristic of expression quantitative trait loci (eQTLs). Complex linkage disequilibrium (LD) between SNPs and causative mutations compounded by sampling errors in test statistics [9] obscures the assignment of a given SNP to a biological function. Because of LD, it is important to expand GWAS SNPs that are obtained from only small subsets of total SNPs (as is generally the case) to obtain all SNPs highly linked to the tested set of SNPs.

While there has been more attention recently to epigenetics in studies of obesity and obesity risk (for example, [1,2,10,11]), many recent GWAS either do not include epigenomic or transcriptomic analyses in evaluating GWAS-derived SNPs or do so with insufficient use of important, publicly available databases (for example, [12,13,14,15,16]). Here, we propose a novel method for prioritizing candidates for regulatory SNPs derived from obesity GWAS by extensively using publicly available epigenomic and transcriptomic data. Our prioritization scheme is focused on SNPs preferentially displaying transcription-enhancing epigenetic marks in adipose tissue and associated with genes preferentially expressed in adipose tissue, in conjunction with a rigorous screening for SNPs that overlap predicted allele-specific transcription factor (TF) binding sites. We focused on adipose tissue due to its direct relationship to obesity [1,2]. We demonstrate that our prioritization method can find a subset of novel, highly plausible regulatory SNPs from 50 obesity GWAS. These SNPs are candidates for modulating enhancers, promoters, or long-distance chromatin interactions affecting expression of protein-coding or long non-coding RNA (lncRNA) genes relevant to adipose biology. Our prioritization method is a valuable tool for rigorous identification of regulatory SNP candidates after obtaining GWAS data and can be easily adopted to studies of other complex disorders.

## 2. Results

### 2.1. Candidate Regulatory SNPs (Tier-1 SNPs) for Obesity Risk Were Prioritized from 50 Obesity-Related GWAS

To identify highly credible candidates for causative regulatory variants from obesity-related GWAS, index SNPs from 50 studies of BMI, WHR_adjBMI_, WC_adjBMI_, or HC_adjBMI_, (Appendix A) were expanded (LD *r*^2^ ≥ 0.8, European population, EUR), and the index or proxy SNPs (I/P SNPs) were prioritized using epigenomics, transcriptomics, and transcription factor binding site (TFBS) predictions (Figure 1). The prioritization depended on the assumptions that many of the regulatory SNPs affecting obesity risk should modulate transcription of genes preferentially expressed in adipose tissue, should directly overlap epigenetic regulatory features seen preferentially in adipose tissue, and should overlap a TFBS that is bound to its TF selectively by only one of the two alleles. We used SAT rather than VAT epigenomics because only SAT genome-wide epigenetic profiles are available for adipose tissue. However, analysis of the GTEx RNA-seq database [17] indicated that 87% of the 2245 genes preferentially expressed in VAT were also preferentially expressed in SAT (ratio of ≥2.0 for VAT or SAT TPM, transcripts per million, versus the median TPM of 35 heterologous tissues).

From the 121,064 I/P SNPs, 331 SNPs (called EnhPro SNPs) were found to overlap a peak of open chromatin in SAT and strong enhancer or strong promoter chromatin as well as a peak of active positive regulatory chromatin (histone H3 lysine-27 acetylation, H3K27ac) preferentially in SAT (in SAT and no more than four of 15 heterologous tissues; Appendix A). Overlap of a candidate regulatory SNP with a peak of open chromatin (DnaseI hypersensitive site, DHS) in a relevant tissue increases the likelihood that a TF binds to the SNP-containing oligonucleotide sequence in that tissue. Enhancer or promoter chromatin determinations were based upon enrichment of histone H3 K4 trimethylation (H3K4me3) and H3K27ac for promoter chromatin and of H3K4me1 and H3K27ac for enhancer chromatin (RoadMap Epigenome Project [8]). We identified the subset of 134 EnhPro SNPs at 57 gene loci that are preferentially expressed in SAT or VAT by examining RNA-seq profiles from poly (A)^+^ RNA (GTEx database [17]). This is an important consideration because a regulatory genetic variant is more likely to be genetically linked to the tissue-associated phenotype if the gene whose expression it modulates is preferentially expressed in a tissue associated with that trait. The definition used for preferential expression was that the ratio of SAT or VAT TPM to the median TPM of 35 heterologous tissues was >2.0 and the SAT or VAT TPM was >2.0. Forty-nine of these genes were preferentially expressed in SAT and 73% of them (36) were also preferentially expressed in VAT while eight genes were preferentially expressed in VAT but not SAT (Appendix A).

We subsequently selected those SNPs with strong predictions of overlap with allele-specific TFBS and without appreciable linkage (LD *r*^2^ < 0.2, EUR) to missense SNPs. Lastly, we verified SAT-preferential expression/epigenetics of the SNP-associated gene in the UCSC Genome Browser [18]. The resulting 47 SNPs are referred to as Tier-1 SNPs and were associated with 14 gene loci (Appendix A).

We retained 7 of these 14 gene loci for further study, which were linked to 18 Tier-1 SNPs, including three that were obtained by imputation (Table 1). Most of the TFs predicted to bind to these 18 Tier-1 SNPs with allele-specificity have some known adipose-related functionality (Appendix A). Seven of the 14 genes were excluded from this study because they were previously examined for GWAS regulatory SNPs (*TBX15* [19]) or because they formed a distinct functional group (extracellular matrix-related genes *COL4A2*, *EFEMP1*, *FBN1*, *NID2*, and *ABLIM3*). In addition, a *HOXC* gene subcluster which had nine Tier-1 SNPs in the regions of *MIR196A2*, *HOXC9*, *HOXC4*, *HOXC5*, and *HOXC6* was also excluded. The *HOXC4/C5/C6* intragenic Tier-1 SNPs are in moderate LD (*r*^2^ = 0.45–0.63) with the *MIR196A2*-overlapping SNP, rs11614913 (Appendix A). This SNP affects, in *cis*, the processing of pre-miR-196a2 [20] and was previously implicated in adipose biology [21,22]. The obesity association of the *HOXC4/C5/C6* Tier-1 SNPs might be due to their linkage with rs11614913, as determined by conditional and joint analysis (COJO [23,24] (Appendix A)).

### 2.2. Limited Power in Finding Best Candidate Regulatory SNPs with Several Commonly Used Statistical Methods

As alternatives to our comparative epigenomic/transcriptomic analysis for obesity SNP prioritization (Figure 1), we tried several popular statistical methods using the largest available obesity GWAS summary data [25], some epigenetic data for SAT [8], and/or SAT eQTLs [17]. First, to identify SNPs and genes with potential causal effects on obesity risk, we used the summary data-based Mendelian randomization with heterogeneity in dependent instruments test (SMR and HEIDI [9]) to prioritize SNPs that have statistically significant potential to affect both the transcript level of a gene and obesity based upon data for SAT eQTLs at a locus [17] and obesity GWAS data. The resulting 58 SNPs that were associated with 65 genes contained no SAT EnhPro SNPs, and only nine genes, including *FAM13A*, showed preferential expression in SAT (Appendix A). Only 10 of these 58 SNPs overlapped a narrow peak of DNaseI hypersensitivity in SAT and only 30 were in a DNaseI hypersensitive region in any of 53 examined tissues or cell types. Expansion of the 58 SNPs using LD *r*^2^ ≥ 0.80 (EUR) yielded 5395 SNPs. Only three of these 5395 SNPs (*FAM13A* SNPs rs11097198, rs13133548, and rs2869949; Table 1 and Appendix A) were Tier-1 SNPs. In contrast, 30 Tier-1 SNPs were derived by our epigenomic/transcriptomic scheme from Pulit et al. GWAS summary data [25].

We also performed colocalization analysis (eCAVIAR [26]) to look for potential causal variants at a given locus that could be causal in both GWAS and eQTL studies among the seven studied gene regions (Table 1). None of our EnhPro/Tier-1 SNPs nor any new credible regulatory SNPs were prioritized by colocalization analysis (Appendix A). In addition, we conducted fine-mapping analysis (Probabilistic Annotation INtegraTOR, PAINTOR [16]) to prioritize potential causal SNPs from the neighborhoods of the above-mentioned seven genes. The fine-mapping analysis of WHR_adjBMI_ that included SAT epigenetic parameters prioritized only two of 18 Tier-1 SNPs (*FAM13A* rs13133548, posterior probability, PP = 1, and *ZBED3-AS1* rs9293708, PP = 0.8; Appendix A). Of the other 55 SNPs prioritized by this fine-mapping analysis (PP ≥ 0.6), only four overlapped enhancer or promoter chromatin and a peak of DNaseI hypersensitivity in SAT (*FAM13A*: rs4544678, rs3775380; *HOXA11*: rs17471520: and *PEMT*: rs8070128). Importantly, four of the seven gene loci had at least one fine-mapping prioritized SNP which was in moderate to high LD (*r*^2^ > 0.4) with a Tier-1 SNP (Table 1 and Appendix A). Although regulatory SNPs can modulate obesity through tissues other than SAT, Tier-1 SNP rs7732130, which as described below was implicated in T2D risk in pancreas by detailed experimental analysis [27,28], was not prioritized by any of the analytical methods that we used (Appendix A).

Because the protocol used in the present study to obtain Tier-1 SNPs (Figure 1) was not meant to be all-inclusive, but rather to give some highly credible risk alleles, we tried to expand it. The last criterion for being a Tier-1 SNP, overlap of a stringently predicted allele-specific TFBS, is limited in power. Five EnhPro SNPs associated with two of the seven genes in Table 1, *FAM13A* and *RP11-392O17.1*, did not meet this last criterion. We utilized COJO analysis [23,24] to look for secondary association signals for the Tier-1 SNPs at neighboring *FAM13A* and *RP11-392O17.1.* This analysis revealed three *FAM13A*-associated EnhPro SNPs (i.e., rs17014602, rs4544678, and rs3775378) that might still be credible obesity risk-associated variants. The *p*-value for the *FAM13A* Tier-1 SNP rs13133548 increased after conditioning on these three EnhPro SNPs (Appendix A). Therefore, our protocol could be extended in the future to include COJO analysis of EnhPro SNPs that did not have predictions of overlapping allele-specific TFBS.

### 2.3. CABLES1, PEMT, and PC Genes Are Associated with Obesity Tier-1 SNPs That May Regulate Alternate Usage of Transcription Start Sites

One or more of the Tier-1 SNPs associated with the genes encoding Cdk5 and ABL1 Enzyme Substrate 1 (*CABLES1*), Phosphatidylethanolamine N-Methyltransferase (*PEMT*), and Pyruvate Carboxylase (*PC*) were located within 0.7 kb of an alternative transcription start site (TSS; Figure 2A and Appendix A). *CABLES1*, *PEMT*, and *PC* are the only genes in their 1-Mb neighborhoods that exhibit preferential expression in SAT (Figure 2 and Appendix A). However, expression of these genes is not specific for adipose tissue, and correspondingly, they have non-adipose tissue-specific functions and generalized functions as well as adipose-related functions [29]. Although *CABLES1* is involved in the regulation of proliferation of pituitary gland cells [30], its expression in SAT or VAT is about 3 times that in the pituitary gland (Appendix A).

There were tissue-specific differences in the usage of alternative TSS for *CABLES1*, *PEMT*, and *PC*, including in SAT versus some other tissues (Appendix A), which might be influenced by Tier-1 SNP-containing enhancer chromatin near one of the TSS. Enhancer chromatin, peaks of H3K27ac, and DNaseI hypersensitivity overlapped some of the Tier-1 SNPs for these genes not only preferentially in SAT but also in SAT-derived mesenchymal stem/stromal cells (SAT-MSC) or adipocytes derived from them in vitro [31] (Table 1, Figure 2B–D; Appendix A). DNA hypomethylated regions, a frequent hallmark of *cis*-acting positive transcription-regulatory elements, overlapped several of *CABLES1* Tier-1 SNPs specifically in SAT-MSC, adipocytes, brain neurons, and the nonneuronal cell fraction of brain (Figure 2E and Appendix A). The brain cell-specific DNA hypomethylation is likely to reflect specific roles for *CABLES1* in the brain [32] and might be related to differences in TSS usage in brain versus adipose (Appendix A). The Tier-1 SNP-containing enhancer chromatin of *CABLES1*, *PEMT*, and *PC* may regulate overall transcription as well as TSS usage. This is illustrated by melanocytes, which had very high overall levels of expression of *CABLES1*, extensive enhancer chromatin and H3K27ac around TSS 2, and much transcription initiation from TSS 2 as well as TSS 1 (Appendix A, data not shown).

The tissue-specific TSS usage for *CABLES1* and *PEMT* results in very different sized polypeptides (Appendix A). However, despite the tissue-specificity for alternate TSS usage for *PC* (*ENSG00000173599*; Appendix A), there is no effect of TSS choice on the encoded polypeptide because the open reading frame (ORF) starts far downstream of all TSS. In addition to the annotated TSS 2, we found a novel, cell type-specific TSS (TSS 3) that is even closer to Tier-1 SNP rs17147932 (0.7 versus 2.5 kb upstream; Appendix A). Only transcription initiation from TSS 1 allows a *PC*-intragenic *MIR3163* gene to be included in the primary transcript (Appendix A). The mature miR-3163 has many documented effects on cell biology partly through modulation of Wnt signaling [33]. SAT and liver share promoter/enhancer chromatin, a H3K27ac peak, and a DHS overlapping rs17147932 at *PC* (Appendix A). The frequency of TSS usage might be modulated by this Tier-1 SNP in the liver, which has the highest expression of *PC*, and in adipose, which has the second highest expression level. Supporting a role of TSS 2 in both the adipose and liver lineages, a binding site in adipocytes was found at the core promoter region of TSS 2 for the adipogenesis-associated TF PPARG [34] and in liver and HepG2 cells at the core promoter regions of TSS 2 and TSS 3 for many transcription initiation-associated proteins (Appendix A).

### 2.4. Only One of the Previously Prioritized FAM13A Obesity-Related Regulatory SNP Candidates Is among the Six Tier-1 FAM13A SNPs

*FAM13A* (*Family with Sequence Similarity 13 Member A*) was associated with six Tier-1 SNPs (Figure 3A, Table 1), four of which were found to be eQTLs in SAT (Appendix A). This gene is implicated in modulating diet-induced obesity, regulating insulin signaling in adipocytes, participating in Wnt signaling, and affecting the risk of certain lung diseases [3,35,36]. We evaluated five previously reported candidate regulatory SNPs (Figure 3A, blue bars and blue font). One, rs2276936, was highlighted by Lin et al. [37] as a likely regulatory SNP from massively parallel reporter assays (MPRA) on bronchial epithelial cells, GWAS data for body fat distribution and blood lipid levels, reporter gene experiments on HepG2 cells, and a CRISPR/Cas-9 mediated ~0.1-kb deletion around this SNP in HepG2 cells. We found that the 0.5-kb region centered around rs2276936 (Figure 3A, blue arrow) did not overlap a peak of DHS (open chromatin) or of H3K27ac in >15 types of cells or tissues, including SAT, liver, HepG2 cells, lung, and esophagus, although the SNP was embedded in a broad region of enhancer chromatin in SAT (Figure 3A–D). Therefore, rs2276936 does not meet the general expectation that a SNP whose allelic state modulates transcription is located at a TF-binding site within (and not just near) a peak of open chromatin and of H3K27ac in a biologically relevant cell type or tissue.

Of the four other previous regulatory candidate SNPs in this region [3,11,14], rs1377290 did not overlap a DHS peak or a H3K27ac peak in SAT and rs9991328 did not overlap a DHS peak in SAT or adipocytes (Figure 3A–D). Another of these previously reported SNPs, rs3822072, lacked specificity for its peak of H3K27ac (present in eight of 15 non-adipose tissues) and was missing a predicted overlapping allele-specific TFBS. The fourth SNP, rs13133548 [15], met all the criteria for a Tier-1 SNP. These five previously described regulatory candidate SNPs are in high or perfect LD (*r*^2^ = 0.78–1.00, EUR) with our four novel Tier-1 SNPs in introns 1 or 2 of the short isoforms of *FAM13A* (Table 1). The last novel Tier-1 SNP, rs7660000, is ~7 kb upstream of TSS 1 in intron 7 of the long *FAM13A* isoform initiated at TSS 2 and located 235 kb upstream of TSS 1 (Appendix A). Only the long FAM13A protein isoform (Appendix A) contains a signaling-associated RhoGAP domain [38]. More transcript is made from *FAM13A* TSS 1 than TSS 2 in SAT, SAT-MSC, and preadipocytes as well as most, but not all cell types (Appendix A). Tier-1 SNPs for *FAM13A* might contribute to different frequencies of use of TSS 1 and TSS 2 as well as tissue-specific differences in expression levels of this gene (Figure 3F).

### 2.5. RP11-392O17.1, an Often-Overlooked SAT-Specific lncRNA Gene, Has Novel Tier-1 SNPs

Of the 14 gene loci associated with Tier-1 SNPs, only *RP11-392O17.1*, *HOXA11*, *MIR196A2/HOXC4*,*5*,*6*,*9*, and *TBX15* displayed a strong bias for expression [17] in SAT versus VAT as well as for SAT versus non-adipose tissues (Figure 4A, Appendix A). As described above, we did not further examine the *HOXC* and *TBX15* loci in this study. For *HOXA11*, a single Tier-1 SNP in the 3′ untranslated region (3′-UTR) near both *HOXA11-AS1* and *HOXA10* was identified (Appendix A). Because it overlaps enhancer chromatin upstream of *HOXA10*, it may regulate *HOXA10* expression. The 3′ location of this SNP raises the alternate possibility that it is involved in post-transcriptional control. In contrast, rs7555150 and rs12025363, the novel Tier-1 SNPs for *RP11-392O17.1*, are 0.6 or 2.3 kb upstream of the TSS in promoter or enhancer chromatin in SAT, which clearly suggests a possible transcription regulatory function. These SNPs are in perfect LD with each other and overlap SAT eQTLs for *RP11-392O17.1* (Appendix A), as reported previously for other nearby SNPs [39]. The *RP11-392O17.1* Tier-1 SNPs display a much stronger association with female rather than male obesity [13], and gender bias was also found for *FAM13A*, *PEMT*, and *ZBED3-AS1* Tier-1 SNPs (Appendix A).

*RP11-392O17.1* is a little-studied lncRNA gene, which has recently been implicated in adipogenesis [40]. The 3-Mb region surrounding this gene has no coding gene with preferential transcription in SAT versus VAT like that of *RP11-392O17.1* (Appendix A). However, the very small and very weakly expressed lncRNA gene *RP11-95P13.2* (SAT TPM, 0.5) 98 kb upstream of *RP11-392O17.1* does show preferential expression in SAT. *RP11-392O17.1* has not yet been designated a RefSeq gene although it is in the Ensembl gene database (*ENSG00000228536*) and displays highly cell type-specific expression in SAT-MSC, SAT, and adult skin fibroblasts (Figure 4B and Appendix A). This gene had transcription-promoting epigenetic marks specifically in SAT, adipocytes, and SAT-MSC in the region of its Tier-1 SNPs, rs7555150, and rs12025363 (Figure 4C–G). Moreover, 13-bp downstream of its TSS, there is a binding site in adipose stem cells for PPARG, the adipose-associated TF (Unibind database [18,41]).

In eight GWAS related to obesity ([42,43] and Appendix A legend), many candidate regulatory SNPs located 11–120 kb upstream of *RP11-392O17.1* were associated with *LYPLAL1* (*Lysophospholipase-like 1*) and overlapped the RefSeq gene structure *LYPLAL1-AS1*/NR_135822.1 (Figure 4H, tan rectangle). The name *LYPLAL1-AS1* is sometimes used in the literature to denote *RP11-392O17.1* or *LYPLAL1-DT*, a lncRNA gene upstream of *LYPLAL1* (Figure 4H and Appendix A) (for example, [40]). There is no evidence for transcription of *LYPLAL1-AS1*/NR_135822.1 in any of >20 examined cell types from the UCSC Genome Browser (Appendix A)*. LYPLAL1* is broadly expressed and located 0.2 Mb further from these SNPs than is its closest downstream gene, *RP11-392O17.1* (Figure 4H). Chromatin state segmentation tracks show the absence of promoter chromatin at the 5′ end of the *LYPLAL1-AS1*/NR_135822.1 in tissues and cell cultures (Appendix A) unlike its presence at the 5′ end of *RP11-392O17.1* in SAT and adult skin fibroblasts (Figure 4D). Importantly, these previously identified obesity GWAS SNPs that other investigators associated with *LYPLAL1* (Appendix A) exhibit moderate LD (*r*^2^ = 0.35 to 0.68, EUR) with Tier-1 SNPs in the promoter region of *RP11-392O17.1*. Therefore, they may have been detected in obesity GWAS only because of their LD with *RP11-392O17.1* promoter-region Tier-1 SNPs.

We tried to test for allele-specific expression of *RP11-392O17.1* in a SAT-MSC cell strain that we identified as heterozygous for both Tier-1 SNPs. Using RT-PCR on total RNA, we amplified two small regions of cDNA not far from the 5′ end of the gene (Figure 4A, lollipops) that contained SNPs in high LD with our Tier-1 SNPs. Sanger DNA sequencing of the PCR products (Appendix A) from the cDNA and from the analogous genomic DNA revealed amplified background sequences that interfered with quantification of the relative expression of the alleles. We were only able to verify transcription of the examined 5′ portions of the gene in SAT-MSC from both alleles, unlike the finding on adipose tissue from reanalysis of RNA-seq data [44].

### 2.6. Two Obesity Tier-1 SNPs Located in ZBED3-AS1 Might Help Control Expression in SAT of Several Genes

We identified Tier-1 SNPs rs7732130 and rs9293708 in the 3′ end of the lncRNA gene *ZBED3-AS1* (Figure 5A). *ZBED3-AS1* might influence expression in SAT of several genes in its 1-Mb neighborhood, including *ZBED3*, which encodes Zinc Finger BED Domain-Containing Protein 3 and is oriented head-to-head to *ZBED3-AS1* with a 0.5 kb overlap. *ZBED3* and *ZBED3-AS1* have been implicated in adipogenesis [45]. The *ZBED3-AS1* Tier-1 SNPs are in low LD with each other (*r*^2^ = 0.2, EUR) and reside 5 kb apart in separate enhancer chromatin segments and DNase hypersensitivity peaks in SAT (Figure 5B–D). One of them, rs9293708, is located 0.6 kb from a binding site for CTCF (CCCTC-binding factor), which can enable long-distance chromatin looping (Figure 5E,F). Despite *ZBED3-AS1* and *ZBED3* sharing a bidirectional promoter, there is appreciable *ZBED3-AS1* RNA in only one of 19 examined cell strains (melanocytes) in contrast to the broad expression of *ZBED3* in cell cultures (Figure 5G and Appendix A). Among tissues, *ZBED3-AS1* and *ZBED3* were preferentially expressed in thyroid, SAT, and VAT, as were *ZBED3-AS1* neighbors *PDE8B* (encoding Phosphodiesterase 8B) and *CRHBP* (encoding Corticotropin Releasing Hormone-Binding Protein; Figure 5H and Appendix A for *CRHBP*). The latter two genes are relevant to obesity. *PDE8B* is involved in insulin signaling in the pancreas and is implicated in controlling lipolysis and the ratio of VAT/total adipose tissue [12,46]. Overexpression of *CRHBP* can affect weight gain in mice [47]. *ZBED3-AS1* was selectively transcribed in vivo in adipocytes as well as in SAT, as seen in a single-nucleus RNA-seq analysis of tissues [48] (Appendix A).

In pancreatic islets, Miguel–Escalada et al. [28] found that a 0.9-kb enhancer chromatin region containing Tier-1 SNP rs7732130 interacts with and upregulates transcription from promoters of *ZBED3-AS1* and the neighboring *PDE8B*, *ZBED3*, *ZBED3-AS1*, *SNORA47*, and *WDR41* genes. The ratio of expression of *ZBED3-AS1* in SAT or VAT versus the median of 35 dissimilar tissues is 4. In contrast, the analogous ratio for pancreas is only 0.5 (Appendix A); however, expression levels may be higher in pancreatic islets for which comparable RNA-seq data are not available. Another distinction between SAT and pancreatic islets is that pancreatic islets lacked the enhancer chromatin in which the second Tier-1 SNP rs9293708 is embedded, although both tissues had enhancer chromatin overlaying rs7732130 (Figure 5A–C; Appendix A).

Chromatin interaction data for this region were not available for adipocytes. However, mapping of long-range chromatin interactions for a lymphoblast cell line (LCL, GM12878) and foreskin fibroblasts [49,50] suggests that regions near both *ZBED3-AS1* SNPs are involved in such interactions and may be at the boundaries of topologically associating domains (TADs). In GM12878, there was evidence for an interaction between an ~60-bp subregion 0.1 kb from rs7732130, which exhibited weak enhancer chromatin, and a repressed promoter of *PDE8B* ~70 kb downstream (promoter capture Hi-C profile, Figure 5F). Foreskin fibroblasts displayed a strong and cell type-specific TAD that extended from a constitutive CTCF-binding site 0.5 kb away from rs9293708 in repressed chromatin to another constitutive CTCF binding site in the active *WDR41* promoter 342 kb away (Figure 5B,E,F and Appendix A). The TAD was seen in Micro-C chromatin interaction profiles, a modified type of Hi-C [50] that is not restricted to interactions using promoter regions as bait. The role of this TAD in foreskin fibroblasts is not clear but the TAD might be needed to insulate the several *PDE8B* promoters from enhancer chromatin seen in skin fibroblasts more than 0.3 Mb upstream (data not shown). Although some of the effects on obesity risk of genes in this neighborhood are probably attributable to pancreas and the thyroid gland (for which epigenome profiles were not available), epigenomics and transcriptomics suggests that the two *ZBED3-AS1* Tier-1 SNPs act through adipose tissue to moderate inherited obesity risk.

## 3. Discussion

We prioritized 47 candidates for obesity-risk regulatory SNPs (Tier-1 SNPs) using a detailed approach involving comparisons of epigenomics and transcriptomics in adipose tissue versus non-adipose tissues. Of the 18 SNPs that were studied in detail, all but two (rs13133548 and rs7732130 [15,28]) had not been previously reported as possible regulatory SNPs. We showed that identifying credible candidates for obesity-risk regulatory SNPs using their SAT epigenetics (because VAT epigenomic databases were not available) and the expression in SAT and VAT of their associated genes usually prioritized genes preferentially expressed in VAT as well as in SAT. Thus, these candidates for regulatory SNPs are likely to be relevant to obesity-related disease susceptibility, which is more commonly associated with VAT [3].

Specific roles in adipose biology have been inferred for the seven gene loci studied in detail (Table 1) [3,5,29,40,45,51,52,53,54,55,56,57]. Remarkably, four of these genes, *CABLES1*, *PEMT*, *PC*, and *FAM13A*, had one or more of their Tier-1 SNPs near an alternate tissue-specific TSS. The choice of alternative well-separated TSS for in vivo transcription of these genes is consequential because it affects either the encoded protein structure (*CABLES1*, *PEMT*, and *FAM13A*) or the expression of a miRNA gene in an isoform-specific intron (Appendix A). Therefore, Tier-1 SNPs for these genes could be involved in modulating the structure of the resulting transcript and not just its quantity in SAT and, thereby, affecting inherited obesity risk.

The Tier-1 SNPs at *CABLES1*, *FAM13A*, *PC*, *PEMT*, *ZBED3-AS1*, and *RP11-392O17.1* are highly plausible candidates for modulating transcription (Figure 6). The genes associated with these SNPs are involved in cell signaling, metabolism, and/or post-transcriptional regulation in ways that could modulate adipose tissue formation or physiology. *CABLES1* encodes a growth suppressor protein that acts as a signaling hub for cell growth [58]. It plays a central role in regulating the cell cycle, cell proliferation, and cancer [59]. Although it has not been previously directly linked to obesity, some of its regulatory interactions are very likely to impact obesity [52,60]. For example, CABLES1 affects CDK5 phosphorylation, which, in turn, controls phosphorylation of the critical adipose-related TF PPARG [51]. FAM13A protein is also implicated in diverse types of signaling [61], including adipocyte insulin signaling [36] and Wnt/beta-catenin signaling, which is crucial for normal development and homeostasis of adipose tissue and other lineages [62,63]. Knockdown of *FAM13A* during in vitro adipogenesis increases expression of several adipocyte markers [11,14], while in vitro overexpression leads to preadipocyte apoptosis [64]. In one study [64], mice with double-knockout of *FAM13A* had a slight increase in SAT, which was dependent on their being fed a high-fat diet. Similarly, in another study [11], male *FAM13A* double-knock out mice had a significant increase in body weight but only when maintained on a high-fat diet [11].

Tier-1 SNP-associated lncRNA genes *RP11-392O17.1* and *ZBED3-AS1* have recently been implicated in adipose biology through interaction with specific proteins or RNAs. In SAT-MSC cells differentiating in vitro to adipocytes, overexpression of part of *RP11-392O17.1* (called *LYPLAL1-AS1* despite the lack of evidence for expression of an overlapping *LYPLAL1-AS1* gene structure; Appendix A) increased expression of adipogenesis markers and fat droplet deposition [40]. This finding was partially attributed to the stabilization of desmoplakin, a desmosome protein, and inhibition of Wnt signaling. The prominence of the active chromatin profiles of *RP11-392O17.1* in SAT-MSC and adipocytes is consistent with expression of this gene playing a role in adipocyte progenitor differentiation as well as in adipocyte function (Figure 4). Nonetheless, in 13 obesity-related GWAS (Appendix A), many candidate risk SNPs upstream of *RP11-392O17.1* were assigned to *LYPLAL1* rather than to the much closer *RP11-392O17.1* (Figure 4H). However, *LYPLAL1* displays no preferential expression in any obesity-relevant tissue, unlike *RP11-392O17.1*. On the basis of eQTLs and limited transcription data, in two previous studies, it was proposed that a few obesity GWAS-derived SNPs that were ascribed to *LYPLAL1* may actually be associated with *RP11-391O17.1* [13,39]. Here, extensive transcription, epigenetic, and genetic data support such reassignment from *LYPLAL1* to *RP11-392O17.1* 0.2 Mb downstream from it, and the conclusion that the previously described SNPs in the *RP11-392O17.1*-upstream intergenic region are proxies for regulatory SNPs in the promoter region of this lncRNA gene.

*ZBED3-AS1*, which contains two intragenic Tier-1 SNPs, rs7732130 and rs9293708 (Figure 5), encodes an RNA that acted as a competitive endogenous RNA (ceRNA, sponge RNA) for miR-513a-5p and miR-381-3p [65]. These miRNAs are implicated in nonalcoholic fatty liver disease [66] or gene dysregulation in VAT in morbidly obese individuals [67]. *ZBED3-AS1* displays increases in its steady-state RNA levels during in vitro adipogenesis, and levels of this RNA correlate with expression of adipogenesis markers in liposuction samples [45]. Moreover, production of *ZBED3-AS1* RNA was reported to be necessary for normal levels of expression of markers of adipogenesis during in vitro differentiation [45]. *ZBED3-AS1* poses an interesting example of a gene with intragenic enhancers that may act in *cis* on nearby genes as well as on itself. The results of Xu et al. [45] suggest that *ZBED3-AS1* also functions in *trans* in SAT-MSC.

Our finding that *ZBED3-AS1* SNP rs7732130 is a Tier-1 SNP prior to reading a report by Miguel–Escalada et al. [28] experimentally validating this SNP’s regulatory function serves as an experimental validation of our scheme for regulatory SNP prioritization. In contrast, an online program for rating candidate regulatory SNPs (RegulomeDB [68]) gave this SNP only a modest probability score (0.38, ranging from 0 to 1, with 1 being most likely to be a regulatory variant). This is probably due to one or more of the following features of RegulomeDB. It does not use transcriptome data. It uses epigenetic data that is not tailored to the tissue of interest and relies on a less stringent and comprehensive tool for assigning allele-specific TFBS overlap (SNP2TFBS, [68]). In their study of T2D GWAS SNPs, Miguel–Escalada et al. [28] documented long-distance chromatin interactions in pancreatic islets between enhancer chromatin containing rs7732130 and promoters of neighboring genes. By epigenome editing they showed that these interactions do significantly potentiate transcription of *ZBED3-AS1* gene neighbors as well as *ZBED3-AS1* itself. Moreover, in reporter gene assays on pancreatic islet cells, a 0.9-kb region harboring this SNP gave allele-specific upregulation [27]. We found that rs7732130 and nearby rs9293708 are obesity GWAS-derived Tier-1 SNPs that overlap a DHS peak, a H3K27ac peak, and enhancer chromatin preferentially in SAT, and for rs7732130, also in pancreatic islets. Our analysis combined with the promoter capture Hi-C mapping and rs7732130-subregion epigenome editing of Miguel–Escalada et al. [28] and rs7732130 allele-specific enhancer transfection assays of Greenwald et al. [27] suggest that rs7732130 modulates obesity risk through adipose tissue as well as through pancreas. In contrast, we propose that rs9293708, which is located in a separate enhancer chromatin region specific to SAT (Figure 5B and Appendix A), does so only through adipose tissue. Given the higher expression of *ZBED3-AS1* in SAT than in pancreas (Appendix A), the influence of rs7732130 on obesity risk might derive more from its effect on expression of this gene in adipose tissue (both SAT and VAT) than in pancreas.

The Tier-1 SNPs of *PC* and *PEMT* might also modulate inherited obesity risk through other tissues as well as through adipose. Positive transcription regulatory marks at the *PC* and *PEMT* intragenic Tier-1 SNPs were seen in both liver and SAT, the tissues that most highly express these genes (Appendix A). These findings suggest that regulatory SNP candidates for obesity risk might assert their effect in more than one tissue. Previous studies indicated that many obesity GWAS regulatory SNPs derive their obesity associations even from brain, a tissue very dissimilar to adipose tissue [10]. The role in brain of such regulatory SNP candidates could be evaluated by our scheme focusing on the epigenomics and transcriptomics of brain instead of adipose tissue.

We demonstrated that a detailed epigenomic/transcriptomic/genetic approach is not only valuable for discovery of novel credible candidates for regulatory SNPs (Tier-1 SNPs) but also for re-evaluating previously reported candidate regulatory SNPs. We also showed that most of our prioritized Tier-1 SNPs were not detected with several standard statistical tools for evaluating GWAS SNPs (SMR and HEIDI [9], PAINTOR [16], and eCAVIAR [26]). Moreover, those statistical tools often gave candidates whose epigenetic features made them unlikely to have a transcription regulatory function because, even when the tools included epigenetic parameters, these parameters may not have been weighted sufficiently. Our findings also can aid future studies of the incompletely understood roles of genes such as *RP11-392O17.1*, *ZBED3-AS1*, *FAM13A*, and *HOXA11* in adipose tissue and other normal or malignant tissues [30,55,69,70,71,72]. Most importantly, comprehensive biochemically-based prioritization of credible candidates for regulatory SNPs from obesity GWAS can greatly facilitate the demanding experiments required to directly verify allele-specific effects on gene expression linked to inherited obesity risk.

## 4. Methods

### 4.1. Obesity GWAS Data

Index SNPs associated with obesity, BMI, WHR_adjBMI_, WC_adjBMI_, and HC_adjBMI_ (*p* < 5 × 10^−8^) were retrieved from 50 studies in the NHGRI-EBI GWAS Catalog [73] (downloaded July 2020; Supplemental Appendix A). We expanded the index SNPs to a set of proxy SNPs (*r*^2^ ≥ 0.8, EUR from the 1000 Genome Project Phase 3 [74] by using PLINK v1.9 [75] (https://www.cog-genomics.org/plink/1.9/; accessed on 1 August 2020) and/or LDlink v3.9 [76] (https://ldlink.nci.nih.gov/; accessed on 1 August 2020). For the genes in Table 1, we also extracted imputed SNPs for BMI and WHR_adjBMI_ [25] (*p* < 5 × 10^−8^) in the 1-Mb region surrounding the genes.

### 4.2. Epigenomics

Chromatin state segmentation data were from the 18-state Roadmap Epigenome model [8] (https://egg2.wustl.edu/roadmap/data/byFileType/chromhmmSegmentations/ChmmModels/core_K27ac/jointModel/final/; accessed on 1 August 2020). Promoter chromatin refers to strong promoter chromatin (state 1), and enhancer chromatin to strong enhancer chromatin (states 3, 8, 9, or 10). In Tables, we visualized these and other data in the UCSC Genome browser (http://genome.UCSC.edu; accessed on 1 October 2021) [18] and slightly modified the color-coding for chromatin state segmentation accordingly to improve clarity. In evaluating promoter versus enhancer designations, note that these occasionally functionally overlap depending on the cell and DNA context [77]. DHS and peaks of H3K27ac were defined as narrowPeaks (Roadmap Project [8], https://egg2.wustl.edu/roadmap/data/byFileType/peaks/consolidatedImputed/narrowPeak/; accessed on 1 July 2020) and checked manually in the UCSC Genome Browser. The 15 non-adipose tissues examined to determine strong promoter/enhancer chromatin and H3K27ac peaks preferentially in SAT versus ≤ 4 non-adipose tissues are given in Appendix A. The long-distance chromatin interactions were from GeneHancer tracks (GeneHancer Track Settings (ucsc.edu) [49]; accessed on 1 October 2021) for the LCL or from the Micro-C track at the UCSC Genome Browser (Hi-C and Micro-C Track Settings (ucsc.edu); accessed on 1 October 2021) for foreskin fibroblasts and embryonic stem cells [41]. Adipocytes are mesenchymal stem cells induced to differentiate in vitro to adipocytes [8].

### 4.3. Transcriptomics

Expression ratios were determined from the TPM of SAT or VAT to that of 35 heterologous tissue types (Appendix A) from the GTEx portal [17]. Preferential expression was defined as this ratio for either SAT or VAT > 2 and with a minimum of TPM > 2 for SAT or VAT. Transcript specific expression and single-nuclei expression were also from GTEx (https://gtexportal.org/; V8; accessed on 1 July 2020). Expression data for cell cultures was from strand-specific RNA-seq of total RNA (CSHL long RNA-seq) and 5′ cap analysis of gene expression (http://genome.ucsc.edu [18]; accessed on 1 October 2021). Preadipocytes were primary cultures from SAT.

### 4.4. Excluding EnhPro SNPs Linked to Missense SNPs and Predicting Allele-Specific TFBS

The EnhPro SNPs were examined for appreciable linkage to missense SNPs (HaploReg v4.1; broadinstitute.org; v4.1; July 2020) at an LD r^2^ > 0.2 (EUR) to eliminate such SNPs from further consideration as candidate transcription regulatory SNPs. Predictions of allele-specific TFBS from the remaining SNPs were using the TRANSFAC v2020.1 program and database (http://gene-regulation.com; accessed on 1 October 2021) with manual curation as previously described [78]. Each TF for a matching TFBS had to have TPM ≥ 2 in SAT or VAT. In addition, we used manual curation to retain only those TRANSFAC TFBS predictions for which all the conserved positions had exact matches to the SNP-containing sequence and for which no more than one base in a partly conserved position had only a partial match (at least 20% as good as the best PWM match). We also looked for evidence from a TF ChIP-seq database [41] for TF binding to oligonucleotide sequences containing our Tier-1 SNPs but no evidence of such binding was found. This is not surprising given the severe limitations on the size of available databases for TF binding in various tissues and cell types.

### 4.5. Statistical Analyses to Look for Potential Causal Regulatory SNPs

For the SMR and HEIDI [9] (https://yanglab.westlake.edu.cn/software/smr/; accessed on 1 September 2020), PAINTOR v3.1 [16] (https://github.com/gkichaev/PAINTOR_V3.0; accessed on 1 May 2021), and eCAVIAR [26] (https://github.com/fhormoz/caviar; accessed on 1 June 2021) analyses, we used the top associated SNPs (*p* < 5 × 10^−8^) from sex-combined WHR_adjBMI_ GWAS summary data [25]. For the SMR and HEIDI analysis, we employed default values of p_eQTL_ < 5 × 10^−8^ for SMR and p_eQTL_ < 1.57 × 10^−3^ for HEIDI, a Bonferroni corrected threshold of p_SMR_ < 2.16 × 10^−6^ and p_HEIDI_ > 0.05. For PAINTOR and eCAVIAR analyses, we only considered significant GWAS SNPs within the 1- to 2-Mb neighborhood of Tier-1 genes in Table 1. For PAINTOR, we integrated overlap of the SNPs with exonic DNA, chromatin segmentation states 1, 3, 8, 9, or 10 (18-state model [8]), or narrow peaks of DHS and H3K27ac in SAT [18]. For colocalization eCAVIAR, we included SNPs with significant eQTLs in SAT located within the 1- to 2-MB neighborhood of the genes in Table 1. For COJO in genome-wide complex trait analysis (GCTA) [23,24] (https://yanglab.westlake.edu.cn/software/gcta/#GWASanalysis; accessed on 1 July 2021), we used the default parameters to evaluate the changes in the association of Tier-1 SNPs with obesity measures by conditioning on any neighboring EnhPro SNPs that lacked a predicted allele-specific TFBS. In addition, we assessed the Tier-1 SNPs for the *HOXC4/C5/C6* subcluster conditional on the *MIR196A2* SNP rs11644913. The reference panel of European individuals (EUR) from the 1000 Genomes Project (phase 3) was used for linkage disequilibrium estimates for all analyses because most GWAS data was for this group.

### 4.6. Transcription from RP11-392O17.1 in SAT-MSC

Total RNA was isolated (https://www.foreivd.com/cell-total-rna-isolation-kit-product/; accessed on 1 March 2021) from low-passage MSC derived from SAT that were >95% positive for CD73, CD90, and CD105 and <2% positive for CD45, CD34, CD11b, and CD19 (www.cellcook.com; accessed on 1 April 2021). First-strand cDNA synthesis (www.uebio.com/productDe_46.html; accessed on 1 May 2021) used ~300 ng of RNA template and included a double-strand DNA-specific DNase in the reaction mixture. PCR (www.uebio.com/productDe_443.html; accessed on 1 May 2021) was just for 35 cycles or for 35 cycles followed by a second round of 25 cycles on a 1:10 dilution of the first-round product.

## 5. Conclusions

A method is presented for prioritizing candidates for GWAS-derived regulatory SNPs by selecting SNPs that overlap multiple epigenetic marks preferentially in a tissue relevant to the GWAS phenotype and that also overlap a predicted allele-specific transcription factor binding site. In addition, the gene associated with the SNP was required to be preferentially expressed in the selected tissue. Using obesity-related GWAS and adipose as the highlighted tissue, 47 SNPs associated with 14 gene loci were prioritized with this method. These prioritized SNPs are highly plausible candidates for influencing inherited obesity risk through allele-specific modulation of transcription levels and, for a few genes, relative usage of multiple alternative promoters. One of the prioritized SNPs was independently examined in two recent experimental studies [27,28]. These studies strongly implicate it in transcription regulation in *cis* in an allele-specific manner. The method described in the present study is an important tool for rigorous prioritization of regulatory SNP candidates that is needed before embarking on demanding experiments to test allelic effects on transcription and can be easily adopted to evaluate transcription-regulatory SNP candidates for other complex disorders.

## Figures and Tables

**Figure 1 ijms-23-01271-f001:**
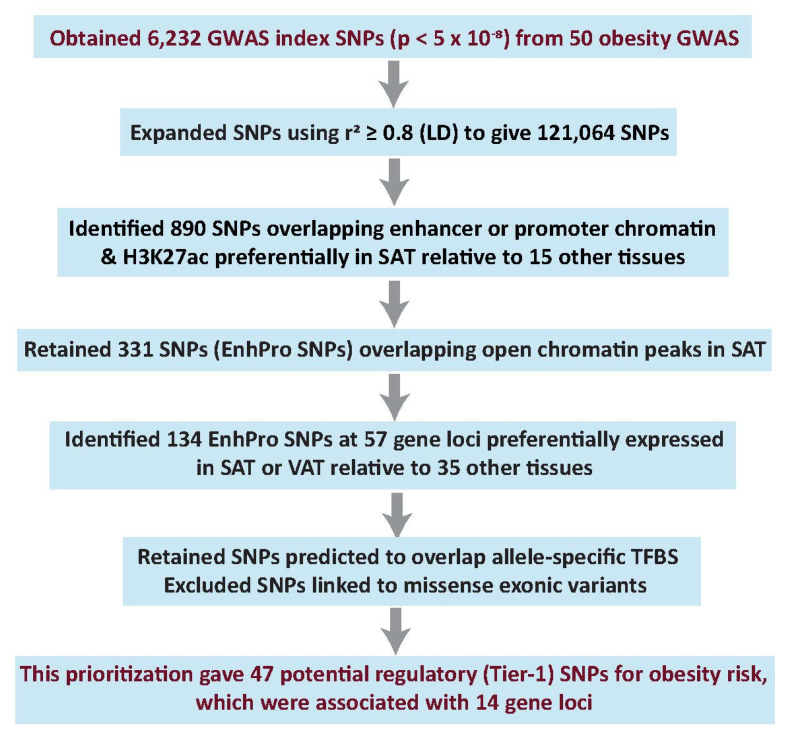
Workflow for prioritizing potential causal regulatory (Tier-1) SNPs from 50 obesity GWAS. Summary of prioritization of plausible regulatory obesity SNPs. LD, linkage disequilibrium; assoc, associated; LD, linkage disequilibrium; H3K27ac, histone H3 lysine-27 acetylation (an epigenetic mark of active enhancers or promoters); SAT, subcutaneous adipose tissue; DHS, DNaseI hypersensitive site (open chromatin); TFBS, transcription factor binding site (s).

**Figure 2 ijms-23-01271-f002:**
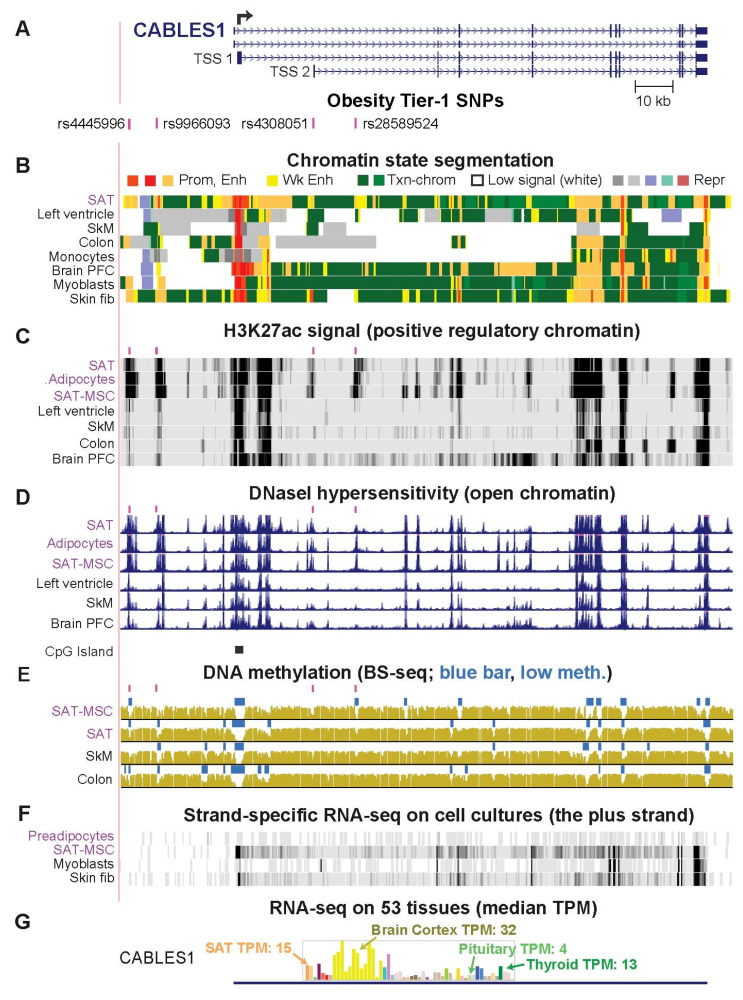
*CABLES1*, which controls cell growth, is associated with Tier-1 SNPs that might regulate alternative usage of its widely separated TSS. (**A**) A 162-kb region containing *CABLES1* and four prioritized regulatory SNP candidates (Tier-1 SNP) for obesity risk (chr18:20,684,550–20,847,013). The broken arrow denotes the most frequently used TSS. (**B**) Strong promoter (Prom) or strong enhancer (Enh) chromatin segments; weak enhancer chromatin (Wk Enh); chromatin with the H3K36me3 mark of actively transcribed regions (Txn-chrom); chromatin with low signals for the assessed histone modifications (low signal); repressed (Repr) chromatin (8). (**C**) H3K27ac enrichment profiles (8). (**D**) Profiles of open chromatin (DNaseI hypersensitivity (8)) and a track for CpG islands, CpG-rich regions (13). (**E**) DNA methylation profiles from whole-genome bisulfite sequencing (BS-seq); blue horizontal bars indicate regions of significantly low DNA methylation (low meth) relative to the rest of the same genome. (**F**) RNA-seq on cell cultures (13); just the plus strand is shown with a vertical viewing range of 0–100. (**G**) Tissue RNA-seq depicted as a bar graph with median transcripts per million (TPM) from hundreds of biological replicates (12). First two orange bars, SAT and VAT (visceral adipose tissue); yellow bars, brain samples; see Appendix A for details. SAT, subcutaneous adipose tissue; SkM, skeletal muscle; PFC, pre-frontal cortex; fib, fibroblasts; MSC, mesenchymal stem/stromal cells; short pink bars in panels (**C**–**E**), indicate positions of Tier-1 SNPs. All tracks were visualized in the UCSC Genome Browser (hg19) and are aligned in this figure and Figure 3, Figure 4 and Figure 5.

**Figure 3 ijms-23-01271-f003:**
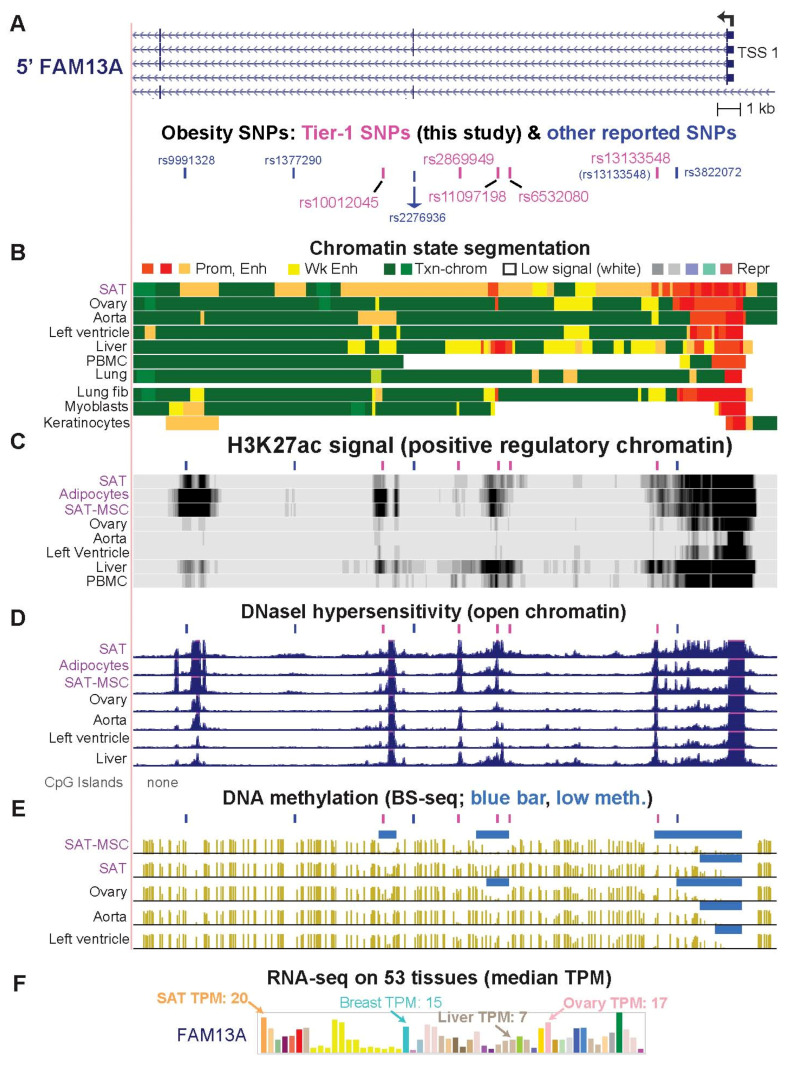
One of the previously highlighted obesity-related SNPs in *FAM13A* (rs13133548) has optimal epigenetic features to be an obesity-risk regulatory SNP. (**A**) A 37-kb region (chr4:89,710,207–89,746,974) overlapping five of the six Tier-1 SNPs (pink bars) as well as four other SNPs that had been described as regulatory SNP candidates from obesity GWAS (blue bars); rs13133548 is the one Tier-1 SNP that was previously described. (**B**–**E**) Chromatin state segmentation, H3K27ac, DHS, and bisulfite-seq profiles as in Figure 2. (**F**) Bar graph of median expression levels for different tissues as in Figure 2. Pink or blue bars above tracks in panels (**C**–**E**), indicate positions of Tier-1 SNPs and previously highlighted candidates for regulatory SNP, respectively. PBMC, peripheral blood mononuclear cells; SAT-MSC, subcutaneous adipose tissue-derived stem/stromal cells; BS-seq, whole genome bisulfite sequencing; meth, methylation; TPM, transcripts per million.

**Figure 4 ijms-23-01271-f004:**
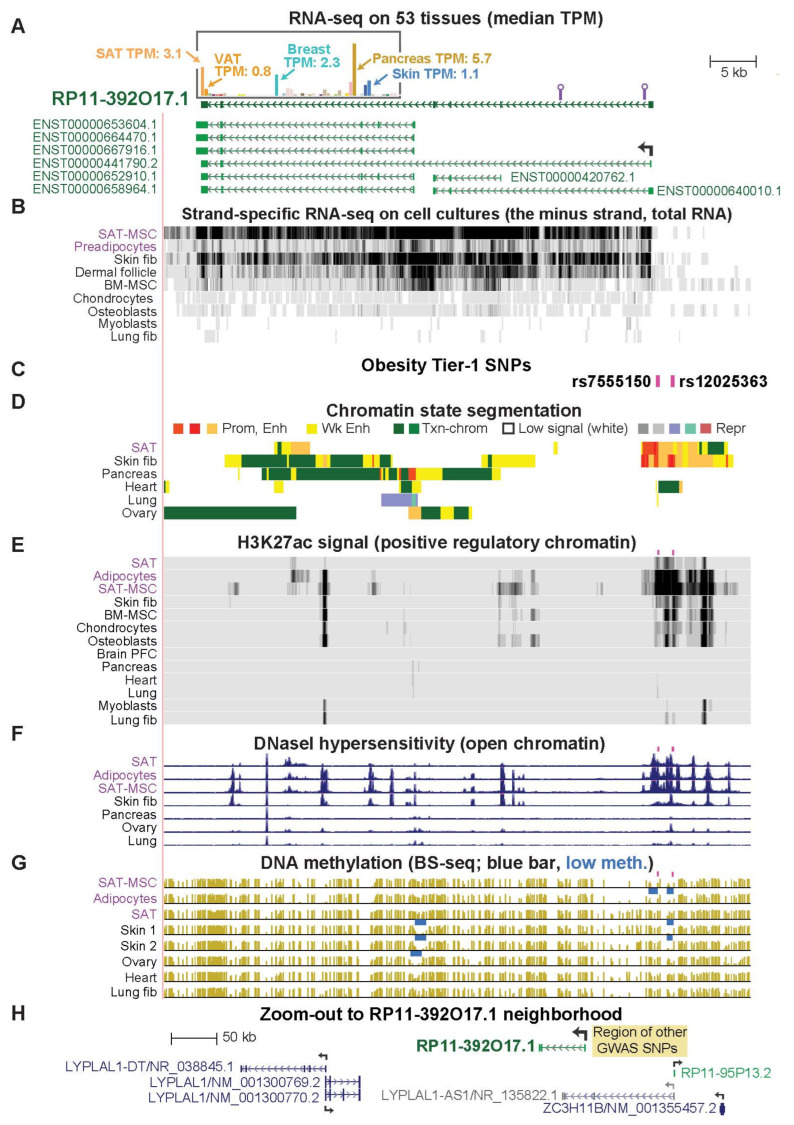
A lncRNA gene, *RP11-392O17.1*, associated with SAT-related and female obesity that is often overlooked in assignments of genes to obesity GWAS SNPs. (**A**) A 65-kb region (chr1:219,578,827–219,643,464) containing *RP11-392O17.1*; above it is the tissue RNA-seq profile (GTEx, determined from poly (A)^+^ RNA) for the gene structure depicted immediately underneath it. The *RP11-392O17.1* isoforms shown are from the latest Ensembl Comprehensive Gene Annotation Set (v104), but this gene is not yet in the RefSeq database. Purple lollipops, regions that we amplified by RT-PCR as described in the text. (**B**) Cell culture RNA-seq (showing the minus strand) on total RNA, vertical viewing range 0–50. (**C**) The location of two Tier-1 SNPs. (**D**–**G**) Chromatin state segmentation, H3K27ac, DHS, and bisulfite-seq, as in the previous figures. There were no CpG islands in the region shown. TPM, transcripts per million; VAT, visceral adipose tissue; SAT-MSC, subcutaneous adipose tissue-derived mesenchymal stem cells; BM-MSC, bone marrow-derived MSC; fib, fibroblasts; BS-seq, whole genome bisulfite sequencing; meth, methylation. (**H**) Annotated genes in the 0.6-Mb region (chr1:219,183,979–219,829,670) that contains *RP11-392O17.1.* The *LYPLAL1-AS1*/NR_135822.1 gene is shown in gray because of the lack of RNA-seq support for this gene structure. Tan rectangle, the region 11–118 kb upstream of *RP11-392O17.1* that was cited in many studies as having obesity GWAS SNPs associated with *LYPLAL1* (Appendix A).

**Figure 5 ijms-23-01271-f005:**
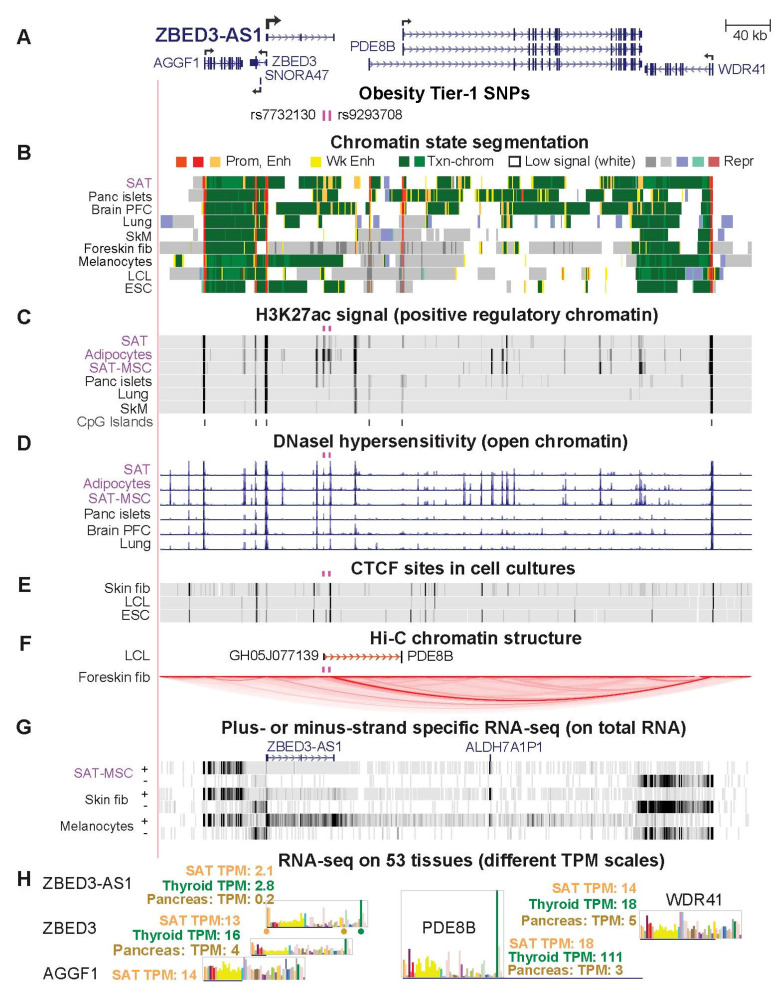
Two Tier-1 SNPs in a lncRNA gene, *ZBED3-AS1*, and neighbors *ZBED3*, *SNORA47*, *PDE8B*, and *WDR41* are near long-distance chromatin looping subregions. (**A**) A 538-kb region (chr5:76,285,728–76,823,921) containing *ZBED3-AS1* and neighboring genes; for clarity, additional isoforms of *ZBED3* and *PDE8B* are shown only in Appendix A. The main TSS for *PDE8B* is indicated by a broken arrow. *CRHBP*, which is referred to in the text but not shown, is 70 kb upstream of *AGGF1*. (**B**–**D**) Chromatin state segmentation, H3K27ac, and DHS as in the previous figures. (**E**) CTCF binding by ChIP-seq profiling of several cell cultures. (**F**) A promoter-capture Hi-C interaction for the lymphoblastoid cell line (LCL) GM12878 and all chromatin interactions in the region detected by Micro-C for foreskin fibroblasts. (**G**) RNA-seq (showing results for transcription from both strands separately) on cell cultures with vertical viewing range 0–30; preadipocytes exhibited no detectable expression of *ZBED3-AS1* (not shown); *ALDH7A1P1* is a pseudogene. (**H**) Tissue RNA-seq shown as a bar graph with dots color-coded to indicate their position in the bar graph for *ZBED3-AS1*. SAT, subcutaneous adipose; panc, pancreatic; PFC, prefrontal cortex; fib, fibroblasts; LCL, lymphoblastoid cell line (GM12878); SAT-MSC, subcutaneous adipose tissue-derived mesenchymal stem/stromal cells; ESC, H1-embryonic stem cells; fib, fibroblasts; TPM, transcripts per million. CpG islands (not shown) overlap all the TSS indicated by broken arrows.

**Figure 6 ijms-23-01271-f006:**
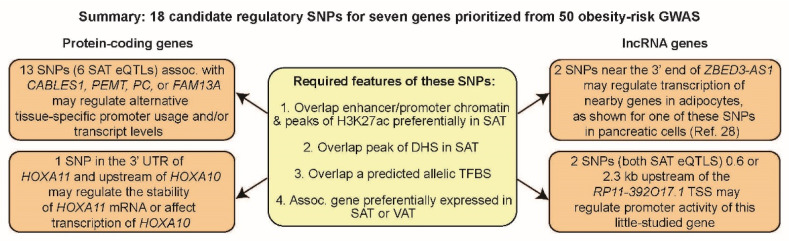
Summary of proposed functions of the 18 obesity-related prioritized candidates for regulatory SNPs (Tier-1 SNPs) and their epigenomic/transcriptomic associations. Assoc., associated; SAT, subcutaneous adipose tissue; VAT, visceral adipose tissue; eQTL, expression quantitative trait loci; 3′ UTR, 3′ untranslated region of mRNA or the corresponding DNA; H3K27ac, histone H3 lysine-27 acetylation, a marker of active enhancer or promoter chromatin; TFBS, transcription factor binding site; Ref. [28], Miguel-Escalada et al. 2019 [28].

**Table 1 ijms-23-01271-t001:** Eighteen prioritized candidates for regulatory obesity-risk variants (Tier-1 SNPs) associated with seven SAT-related genes ^a^.

Gene	Tier-1 SNP for Obesity Association	Index/Proxy (I/P) or Imputed (Imp) SNP	Alleles (Ref or Alt)	Obesity Increasing Allele (Freq EUR) ^b^	Chromatin State at the SNP in SAT ^c^	Gene Location of SNP Relative to Gene	Predicted TF Binding ^d^		Type of Additional Statistical Analysis That Predicts Assocn. ^e^
							Ref Allele	Alt Allele	
CABLES1	rs4445996	Imp	A/G	G (0.91)	Str enh	5′ to TSS 1	DNTTIP1	MAX	None
“	rs9966093	Imp	C/T	T (0.91)	Str enh	5′ to TSS 1	ZXDA, ZXDB	RXRA	None
“	rs4308051 *^,†^	I/P & Imp	T/G	G (0.79)	Str enh	Near TSS 2	BHLHE40/41	KLF4	None
“	rs28589524 *^,†^	I/P	G/A	NA (0.79)	Str enh	Near TSS 2	NFKB1, RELA	SMAD2	None
FAM13A	rs10012045 *^,†^	I/P	C/T	NA (0.56)	Str enh	Intron 2 ^f^	EPAS1	None	None
“	rs2869949 *^,†^	I/P	A/G	NA (0.51)	Str enh	Intron 1	ZNF333	None	SMR&HEIDI
“	rs11097198 *^,†^	I/P	C/T	NA (0.51)	Str enh	Intron 1	STAT1/3/4/5B	None	SMR&HEIDI
“	rs6532080 *^,†^	I/P	A/C	NA (0.56)	Str enh	Intron 1	SOX4	None	None
“	rs13133548 *^,†^	I/P & Imp	G/A	A (0.48)	Str enh/prom	Intron 1	None	POU3F3	PAINTOR; SMR&HEIDI
“	rs7660000	I/P & Imp	C/T	C (0.70)	Str enh	5′ to TSS 1	ZBTB44	None	None
HOXA11	rs530375207	I/P	T/A	NA (0.91)	Str enh	3′-UTR	ZNF254	PRRX2, DLX5	None
PC	rs17147932	I/P & Imp	G/T	T (0.05)	Str enh	Near TSS 2	KLF2/6, SP2/3	None	None
PEMT	rs8078513 *	I/P & Imp	C/A	A (0.07)	Str enh	End of intron 2	SMAD3	None	None
“	rs8070432 *	Imp	T/C	C (0.07)	Str enh	Near TSS 3	None	RUNX1	None
RP11-392O17.1	rs7555150 *^,†^	I/P & Imp	A/G	G (0.61)	Str enh/prom	Near prom	None	NFIB	None
“	rs12025363 *^,†^	I/P & Imp	G/A	A (0.61)	Str enh	Prom region	GTF2IRD1	None	None
ZBED3-AS1	rs7732130	I/P & Imp	G/A	G (0.33)	Str enh	Near 3′ end	RFX7	None	None
“	rs9293708^†^	I/P & Imp	C/T	C (0.43)	Str enh	Near 3′ end	SOX18	None	PAINTOR

^a^ Tier-1 SNPs were obtained from 50 GWAS (Figure 1); SAT, subcutaneous adipose tissue. ^b^ Obesity-increasing allele was from WHR_adjBMI_ summary data by Pulit et al. [25]; freq EUR, frequency in the European population; NA, not available (Ref allele frequency in EUR is given instead). ^c^ Str prom, strong promoter chromatin; Str enh, strong enhancer chromatin from Roadmap database 18-state chromatin segmentation maps; Str enh/prom, strong enhancer or promoter chromatin. ^d^ Transcription factors (TFs) predicted by stringent criteria to bind to SNP-containing sequences in an allele-specific manner. ^e^ The indicated Tier-1 SNP was prioritized by SMR & HEIDI or PAINTOR programs (see Results 2.2); assocn, association. ^f^ Intron nomenclature for *FAM13A* is for the main isoform generated from transcription initiation at the proximal TSS. *, the Tier-1 SNP is in LD r^2^ > 0.75 (EUR) with other Tier-1 SNPs associated with the same gene; other multiple Tier-1 SNPs for a gene are in low LD with each other (r^2^ < 0.3). †, the Tier-1 SNP has LD r^2^ ≥ 0.4 (EUR) with 12 prioritzed plausible causual SNPs (PP ≥ 0.6) from PAINTOR WHR_adjBMI_ with epigenetic annotations. Ref allele, Reference allele; Alt allele, alternate to the Ref allele; 3′-UTR, 3′ untranslated region of mRNA or the corresponding DNA.

## Data Availability

All data are publicly available at the URLs indicated in Methods.

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
