# Peer review of "Epigenomic and Transcriptomic Prioritization of Candidate Obesity-Risk Regulatory GWAS SNPs"

_ijms, 2022, doi:10.3390/ijms23031271_

Round 1
Reviewer 1 Report
Zhang et a. have submitted manuscript entitled "Epigenomic and Transcriptomic Prioritization of Candidate 2 Obesity-Risk Regulatory GWAS SNPs" The manucript is really interesting and provided some novel information about the genetic basis of obesity. My some spoecifc comments are below:
Abstract: 1. The introductory lines are a bit more, I would suggest to make it lesser and focusing on the result and finding parts more.
2. Avoid acryonyms i.e. VS.
Introduction:1.
Sentence on line 35-36 "Adipose tissue expansion involves 35
an increase in adipocytes and/or an expansion of the size of adipocytes" is bit confusing, I would prefer to make it more clear.
2. "These anthropomorphic measures are 51 the most frequent ones used in genome-wide association studies of the genetic risk of obe-52 sity (obesity GWAS)." why? do you have any specific reason?
3. Para from 69-86; It seems too lengthy and reader may loss the path, it is suggested to rephrase with to the point approach.
Results:
- Figure 1. the given schematic, to my eye is a bit blur, also the fonts size is also small, same i felt for other figures too.
- Sometimes I felt that subheadings are too long e.g. "Only one of the previously prioritized FAM13A candidate regulatory SNPs for obesity risk 265 is among the six Tier-1 SNPs identified for this gene in the present study"
Author Response
We thank the reviewers for their helpful comments. We have followed all their suggestions and answered all their questions as shown below in the underlined text.
Zhang et a. have submitted manuscript entitled "Epigenomic and Transcriptomic Prioritization of Candidate 2 Obesity-Risk Regulatory GWAS SNPs" The manucript is really interesting and provided some novel information about the genetic basis of obesity. My some spoecifc comments are below:
Abstract: 1. The introductory lines are a bit more, I would suggest to make it lesser and focusing on the result and finding parts more.
Response: We have shortened the second sentence of the Introduction. The first sentence is already short and needed for the reader unfamiliar with obesity-related GWAS.
- Avoid acryonyms i.e. VS.
Response: Done
Introduction:1.
Sentence on line 35-36 "Adipose tissue expansion involves 35
an increase in adipocytes and/or an expansion of the size of adipocytes" is bit confusing, I would prefer to make it more clear.
Response: We clarified the sentence as follows.
Adipose tissue expansion involves an increase in the number of adipocytes and/or in the size of adipocytes.
- "These anthropomorphic measures are 51 the most frequent ones used in genome-wide association studies of the genetic risk of obe-52 sity (obesity GWAS)." why? do you have any specific reason?
Response: We added the last clause to the sentence.
These anthropomorphic measures are the most frequent ones used in genome-wide association studies of the genetic risk of obesity (obesity GWAS) because of their ease of determination and, for the last three measures, their health-related emphasis on the distribution of fat depots [3].
- Para from 69-86; It seems too lengthy and reader may loss the path, it is suggested to rephrase with to the point approach. Response: Done.
Results:
- Figure 1. the given schematic, to my eye is a bit blur, also the fonts size is also small, same i felt for other figures too. Response: Fonts sizes have been increased in all Figures. In addition, we made Figure 1 shorter with clearer summaries of the method.
- Sometimes I felt that subheadings are too long e.g. "Only one of the previously prioritized FAM13A candidate regulatory SNPs for obesity risk 265 is among the six Tier-1 SNPs identified for this gene in the present study" Response: We have shortened three of the subheadings.
Reviewer 2 Report
The article is well written and justified through suitable evaluation parameters and references. Though it contains sufficient novelty to be accepted for publication, but still minor modifications and suggestions are recommended to improve the quality of the manuscript. I suggest:
- The aim of the study should be clearly defined.
- Is the distribution of selected single nucleotide polymorphisms (SNPs) in obese subjects different from that in healthy subjects?
- Line 470-472: please explain the relationship described.
- Line 515: please give more details on the studies referred to by the authors.
- Please add a subsection "Conclusions".
- What statistical program was used for statistical analysis.
- Improve the quality of Figure 1.
- Standardize the placement of Figures 2-5 in the text.
- Complete and improve justification of "Abbreviations".
- Correct the article according to the instructions for authors (including the "References" section).
Conclusion
Overall, the manuscript could be considered as scientific rigor and seems able to add in existing scientific knowledge. Therefore, I recommend the Acceptance of the manuscript with minor modifications on above mentioned suggestions and comments.
Author Response
We thank the reviewers for their helpful comments. We have followed all their suggestions and answered all their questions as shown below in the underlined text.
The article is well written and justified through suitable evaluation parameters and references. Though it contains sufficient novelty to be accepted for publication, but still minor modifications and suggestions are recommended to improve the quality of the manuscript. I suggest:
- The aim of the study should be clearly defined. Response: We have done so now in the first two sentences of the last paragraph of the Introduction (Lines 72 – 76).
- Is the distribution of selected single nucleotide polymorphisms (SNPs) in obese subjects different from that in healthy subjects?
Response: In GWAS, either linear or logistic regression models are used to test the association between the genetic variants and the phenotype of interest, depending on whether the phenotype values are continuous (e.g., BMI) or binary (e.g., obese vs healthy or non-obese). If a SNP's association reached the genome-wide significant threshold set by the individual GWAS (commonly used threshold is p < 5x10-8), then this SNP is considered to be differently distributed in, for example, high vs low BMI subjects, obese vs healthy subjects, or obese vs non-obese subjects. In our study, we obtained the index or proxy SNPs that reached the genome-wide significance level of p < 5x10-8. We also found that the 18 Tier-1 SNPs were significant in the largest obesity GWAS by Pulit et al. (Suppl Table S4).
- Line 470-472: please explain the relationship described.
Response: Done.
In one study [65], mice with double-knockout of FAM13A had only a slight increase in SAT, which was dependent on their being fed a high-fat diet. Similarly, in another study [11], male FAM13A double-knock out mice had a significant increase in body weight but only when maintained on a high-fat diet [11].
- Line 515: please give more details on the studies referred to by the authors.
Response: Done
Our analysis combined with the promoter capture Hi-C mapping and rs7732130-subregion epigenome editing of Miguel-Escalada et al. [28] and rs7732130 allele-specific enhancer transfection assays of Greenwald et al. [27] suggest that both rs7732130 and rs9293708 modulate obesity risk through adipose tissue and pancreas. In contrast, we propose that rs9293708, located in a separate enhancer chromatin region specific to SAT (Figure S12B and C), does so only through adipose tissue.
- Please add a subsection "Conclusions". Response: Done
- What statistical program was used for statistical analysis. Response: We previously had only the references to the statistical programs cited in the Methods section. We have added the URLs for the statistical programs to this part of Methods.
- Improve the quality of Figure 1. Response: Done
- Standardize the placement of Figures 2-5 in the text. Response: Done
- Complete and improve justification of "Abbreviations". Response: Done
- Correct the article according to the instructions for authors (including the "References" section). Response: Done
Conclusion
Overall, the manuscript could be considered as scientific rigor and seems able to add in existing scientific knowledge. Therefore, I recommend the Acceptance of the manuscript with minor modifications on above mentioned suggestions and comments.